# Fair Adaptive Experiments

**Waverly Wei**
Division of Biostatistics
University of California, Berkeley
linqing_wei@berkeley.edu

**Xinwei Ma**
Department of Economics
University of California, San Diego
x1ma@ucsd.edu

**Jingshen Wang** *
Division of Biostatistics
University of California, Berkeley
jingshenwang@berkeley.edu

## Abstract

Randomized experiments have been the gold standard for assessing the effectiveness of a treatment, policy, or intervention, spanning various fields, including social sciences, biomedical studies, and e-commerce. The classical complete randomization approach assigns treatments based on a pre-specified probability and may lead to inefficient use of data. Adaptive experiments improve upon complete randomization by sequentially learning and updating treatment assignment probabilities using accrued evidence during the experiment. Hence, they can help achieve efficient data use and higher estimation efficiency. However, their application can also raise fairness and equity concerns, as assignment probabilities may vary drastically across groups of participants. Furthermore, when treatment is expected to be extremely beneficial to certain groups of participants, it is more appropriate to expose many of these participants to favorable treatment. In response to these challenges, we propose a fair adaptive experiment strategy that simultaneously enhances data use efficiency, achieves an "envy-free" treatment assignment guarantee, and improves the overall welfare of participants. An important feature of our proposed strategy is that we do not impose parametric modeling assumptions on the outcome variables, making it more versatile and applicable to a wider array of applications. Through our theoretical investigation, we characterize the convergence rate of the estimated treatment effects and the associated standard deviations at the group level and further prove that our adaptive treatment assignment algorithm, despite not having a closed-form expression, approaches the optimal allocation rule asymptotically. Our proof strategy takes into account the fact that the allocation decisions in our design depend on sequentially accumulated data, which poses a significant challenge in characterizing the properties and conducting statistical inference of our method. We further provide simulation evidence and two synthetic data studies to showcase the performance of our fair adaptive experiment strategy.

## 1 Introduction

### 1.1 Motivation and contribution

Randomized experiments are considered gold standards for evaluating the effectiveness of public policies, medical treatments, or advertising strategies [32, 34, 35]. They involve randomly assigning

---

*Corresponding author

37th Conference on Neural Information Processing Systems (NeurIPS 2023).

participants to different treatment groups, allowing for rigorous causal conclusions and robust evidence for decision-making and policy implementation. However, classical randomized experiments, which maintain fixed treatment assignment probabilities, often do not optimize data utilization. This limitation is problematic due to the high costs associated with conducting such experiments. Consequently, maximizing information gain is crucial, but classical randomized experiments do not prioritize this objective [25, 46].

Compared to classical randomized experiments, adaptive experiments provide enhanced information gain and improved statistical efficiency. As a result, adaptive experiments have gained popularity in diverse domains such as field experiments, online A/B testing, and clinical trials [26, 56, 63, 64]. The information gain of adaptive experiments stems from their ability to iteratively adjust treatment allocations based on refined knowledge obtained from accumulated data during the experiment. This iterative process often favors the treatment arm that offers more informative or beneficial outcomes, maximizing the information gained from each participant and optimizing the overall statistical efficiency of the experiment [45]. Moreover, adaptive experiments, thanks to their adept utilization of data resources, often exhibit greater statistical testing power when practitioners employ the collected data to assess the null hypothesis of zero treatment effect upon experiment completion [14].

Despite their appealing benefits in improving data use efficiency and boosting statistical power, adaptive experiments potentially bring fairness concerns in applications. This issue is neither sufficiently explored nor fully addressed in the existing literature. Below, we shall concretely discuss the fairness concerns under a scenario where the study population can be divided into distinct groups based on demographic information or biomarkers — a scenario frequently encountered in field experiments or clinical trials. The *first* fairness concern in adaptive experiments arises when there are significant disparities in treatment allocations among different participant groups [14]. This is because when the treatment is potentially beneficial, it is crucial to ensure a fair chance for each group to receive beneficial treatment. Similarly, it is important to avoid disproportionately burdening any specific group with unfavorable treatment. However, conventional adaptive experiments prioritizing efficiency gains may inadvertently result in unfair treatment allocations. For example, if the outcome of a particular group of participants exhibits a higher variance in response to the treatment, more participants in the group will be allocated to the treatment arm. Consequently, this group would have a significantly higher treatment assignment probability than the others, regardless of the sign and magnitude of the treatment effect. This may lead to an unfair allocation of treatments among participants. Completely randomized experiments with fixed one-half treatment assignments would avoid this challenge, but they suffer from information loss. The *second* fairness concern arises when the adaptive treatment allocation does not adequately account for the overall welfare of experimental participants. This is crucial as a fair experiment is expected to not only assign a large proportion of participants to a beneficial treatment arm but also assign a small proportion of participants to a harmful treatment to avoid adverse effects.

There are evident challenges in addressing fairness concerns while optimizing information gain in adaptive experiments due to the potential trade-off among fairness concerns, welfare improvement, and information gain. For example, if most of the participants are assigned to the beneficial treatment to maximize welfare, then there would be insufficient sample size in the control arm, resulting in imprecisely estimated treatment effect and hence reduced statistical efficiency for conducting inference. To overcome these challenges, we propose a fair adaptive experimental design strategy that balances competing objectives: improving fairness, enhancing overall welfare, and gaining efficiency. By rigorously demonstrating the effectiveness of our approach and providing statistical guarantees, we offer a practical solution that is both grounded in theory and reconciles fairness concerns with the requirement for robust information gain in adaptive experiments. Our contributions can be summarized as follows:

*First*, in comparison to existing adaptive experiments, our proposed strategy integrates fairness and welfare considerations while optimizing information gain. As a result, the treatment allocation probability generated by our method avoids extreme values and exhibits minimal necessary variations across different groups. These desirable characteristics are supported by our simulation studies and empirical illustration using synthetic data. It is important to note that due to the additional constraints of welfare and fairness, the optimal treatment allocation probability does not have a closed-form expression, which brings additional technical challenges to studying the theoretical properties of our design. Despite this challenge, we demonstrate that the constructed treatment allocation rule for each group of our design converges to its oracle counterpart (Theorem 2). This implies that our proposed

designs in Section 2.3, despite not relying on prior knowledge about the underlying data distribution before the start of the experiment, can allocate treatments in a similar manner to a scenario where perfect knowledge of the data distribution is available.

*Second*, we do not impose any specific parametric modeling assumptions on the outcomes beyond mild moment conditions. We instead estimate the mean and variance of potential outcomes at the group level, which are further incorporated into our algorithm. The nonparametric nature of our procedure delivers an efficient and accurate estimation of the average treatment effect. As an important theoretical contribution, we prove that those group-level estimates are asymptotically consistent (Theorem 1).

*Third*, our theoretical framework addresses the challenges and complexities associated with adaptive experiment design, where data are sequentially accumulated, and treatment allocation decisions are adaptively revised, resulting in non-independently and non-identically distributed data. By leveraging the martingale methods, we demonstrate that the estimate of the average treatment effect is consistent and asymptotically normally distributed (Theorems 1 and 3). An important methodological and practical innovation of our framework is that it does not require the number of participants enrolled in the first stage to be proportional to the overall sample size. This flexibility allows researchers to allocate more participants in later stages of the experiment, enabling a truly adaptive approach to experiment design and implementation. This innovation has significant implications for the methodology and practical application of adaptive experiments.

## 1.2 Related literature

Our proposed fair adaptive experiment strategy has a natural connection with the response adaptive randomization (RAR) design literature. The early work develops the randomized play-the-winner rule in clinical trial settings based on urn models [46, 48, 60]. Theoretical properties of urn models are investigated in [6] and [29]. Another conventional response adaptive design is the doubly adaptive biased coin (DBCD) design [15, 26, 27, 53]. However, to our best knowledge, many existing works on response adaptive designs do not take fair treatment allocations into account [24, 47]. An insightful work in [33] proposes an efficient RAR design to minimize the variance of the average treatment effects and discusses some directions for fair experimental design. Compared with [33], our method does not require estimating outcome models, which can be challenging in the presence of correlated data in RAR designs. In addition, our design centers around "group" fairness, aiming to enhance participants' well-being while avoiding extra fairness complications among distinct individuals. Furthermore, RAR designs that further incorporate covariate information are known as covariate-adjusted response adaptive (CARA) designs [7, 9, 37, 49, 57, 71, 72]. Some early work proposes to balance covariates based on the biased coin design [44, 69]. Later work considers CARA designs that account for both efficiency and ethics [28] and extends the CARA design framework to incorporate nonparametric estimates of the conditional response function [1]. It is worth mentioning that another strand of literature focuses on ethical designs using Bayesian frameworks. Some recent work proposes to use the Gittins index to improve participants' welfare [55, 58]. A later work develops a Bayesian ethical design to further improve statistical power [62]. Some other ethical designs are discussed in [18, 52, 68].

Our manuscript also relates to the literature on semiparametric efficiency and treatment effect estimation [20, 42, 54]. Our algorithm adaptively allocates participants to treatment and control arms, with the aim of not only minimizing the variance of the estimated average treatment effect but also incorporating constraints on fairness and welfare. There is also a large literature on efficient estimation of treatment effects and, more broadly, on estimation and statistical inference in semiparametric models. See, for example, [8, 10, 11, 12, 16, 22, 38, 39, 59, 61] and references therein. Our algorithm takes the group structure as given. Another strand of literature studies stratified randomization. Some recent contributions in this area include [5, 51].

Lastly, our proposed design is connected to the multi-armed bandit (MAB) literature. [50] studies the trade-off between regret and statistical estimation efficiency by formulating a minimax multi-objective optimization problem and proposing an effective Pareto optimal MAB experiment. They provide insightful theoretical results on the sufficient and necessary conditions for the Pareto optimal solutions. Our procedure attains the minimax lower bound for fair experiment design problems. Our work has a different focus on uncovering the underlying causal effect by providing an adaptive procedure for efficient treatment effect estimation while incorporating fairness and welfare consid-

erations. Furthermore, in the theoretical investigations, we focus on the asymptotic normality of the proposed estimator and its variance estimator, which enables valid statistical inference. Our work broadly connects with fair contextual bandit literature [13, 17, 19, 30, 40, 43]. [65] and [66] propose algorithms under subpopulation fairness and equity requirements for the tasks of best arm identification and ranking and selection. The work in [31] characterizes fairness under the contextual bandit setting by bridging the fair contextual bandit problems with "Knows What It Knowns" learning. While [31] defines fairness metric on the individual level, we focus on group-level fairness and further incorporate a welfare constraint.

## 2 Fair adaptive experiment

### 2.1 Problem formulation and notation

In this section, we formalize our adaptive experiment framework and introduce necessary notations. In adaptive experiments, participants are sequentially enrolled across $T$ stages. We denote the total number of enrolled participants as $N = \sum_{t=1}^{T} n_t$, where $n_t$ is the number of participants in Stage $t$, $t = 1, \ldots, T$. In line with the existing literature [24, 25, 28], we assume $T \to \infty$, and $n_t$ is small relative to the overall sample size $N$, meaning that we have many opportunities to revise the treatment allocation rule during the experiment (see Assumption 3 below). At Stage $t$, we denote participant $i$'s treatment assignment status as $D_{it} \in \{0, 1\}$, $i = 1, \ldots, n_t$, with $D_{it} = 1$ being the treatment arm and $D_{it} = 0$ being the control arm. Denote participant $i$'s covariate information as $X_{it} \in \mathbb{R}^p$ and the observed outcome as $Y_{it} \in \mathbb{R}$.

Next, we quantify causal effects under the Neyman-Rubin potential outcomes framework. Define $Y_{it}(d)$ as the potential outcome we would have observed if participant $i$ receives treatment $d$ at Stage $t$, $d \in \{0, 1\}$. The observed outcome can be written as

$$Y_{it} = D_{it} Y_{it}(1) + (1 - D_{it}) Y_{it}(0), \quad i = 1, \ldots, n_t, \ t = 1, \ldots, T. \tag{1}$$

In accordance with classical adaptive experiments literature, we assume that the outcomes are observed without delay, and their underlying distributions do not shift over time [25]. The average treatment effect (ATE) is the mean difference between the two potential outcomes:

$$\tau = \mathbb{E}[Y_{it}(1) - Y_{it}(0)]. \tag{2}$$

In our proposed fair adaptive experiment strategy, to protect the participant's welfare (more discussions in Section 2.2), we also consider the group-level treatment effects. We assume the study population can be partitioned based on demographics or biomarkers, which is frequently seen in clinical settings or social science studies [3, 36, 67]. More concretely, by dividing the sample space $\mathcal{X}$ of the covariate $X_{it}$ into $m$ non-overlapping regions, denoted as $\{\mathcal{S}_j\}_{j=1}^{m}$, we define the treatment effect in each group as

$$\tau_j = \mathbb{E}[Y_{it}(1) - Y_{it}(0) | X_{it} \in \mathcal{S}_j], \ j = 1, \ldots, m. \tag{3}$$

We further denote the total number of participants enrolled in the group $j$ as $N_j = \sum_{t=1}^{T} n_{tj}$.

In adaptive experiments, as we aim to adaptively revise the treatment assignment probabilities based on the evidence accrued during the experiment to meet our fairness and efficiency goals, we define treatment assignment probability (or propensity scores) for participants in groups $j$ at stage $T$ as

$$e_{tj} := \mathbb{P}(D_{it} = 1 | X_{it} \in \mathcal{S}_j, \text{ history up to time } t - 1), \ t = 1, \ldots, T, \ j = 1, \ldots, m. \tag{4}$$

The goal of our experiment is to dynamically revise $e_{tj}$ for efficiency improvement, fairness guarantee, and welfare enhancement.

### 2.2 Design objective in an oracle setting

Classical adaptive experiments, aimed at reducing variance (or, equivalently, efficiency improvement), often assign treatment using Neyman allocation for participants in each group, that is

$$e_{j,\text{Neyman}}^* = \frac{\sigma_j(1)}{\sigma_j(1) + \sigma_j(0)}, \quad j = 1, \ldots, m, \tag{5}$$

where $\sigma_j^2(d) = \mathbb{V}[Y_{it}(d)|X_{it} \in \mathcal{S}_j]$, $d \in \{0, 1\}$, denotes the variance of the potential outcome under treatment arm $d$ in group $j$. Although Neyman allocation improves the estimation efficiency of the ATE, it brings two critical fairness concerns. *First*, different groups of participants may have substantially different probabilities of receiving treatments. The form of Eq (5) implies that the treatment assignment probabilities solely rely on group-level variances under different arms. More specifically, the group of participants with a larger variance under the treatment arm will have a higher probability of being treated, which may lead to disproportionate treatment allocations across different groups. *Second*, some participants' welfare could be harmed under the adaptive experiment strategy in Eq (5). To see this, assume a group of participants exhibits a large variance yet rather negative responses under the treatment arm. However, more participants in this group will be assigned to the treatment arm to improve the estimation efficiency of ATE despite the impairment of those participants' welfare.

To address fairness concerns and facilitate the introduction of our experimental goals, we begin with an infeasible "oracle" setting, where we possess knowledge of the true underlying data distribution before the experiment begins. Since adaptive experiments naturally allow for sequential learning of unknown parameters and adjustment of treatment allocations during the experiment, we will present our adaptive experimental design strategy in the following section (Section 2.3), which attains the same theoretical guarantee for estimating the ATE as in the oracle setting (see Theorems 2 and 3 for justification).

In the oracle setting, given we have perfect knowledge of the underlying data distribution (thus $\tau_j$ and $\sigma_j^2(d)$ are known to us), our goal is to find optimal treatment allocations $\boldsymbol{e}^* = (e_1^*, \ldots, e_m^*)^\mathsf{T}$ that solve the following optimization problem:

**Problem A**

$$\min_{\boldsymbol{e}} \ \sum_{j=1}^m p_j \Big( \frac{\sigma_j^2(1)}{e_j} + \frac{\sigma_j^2(0)}{1 - e_j} \Big), \qquad\qquad \leftarrow \text{ Improve estimation efficiency for the ATE}$$

$$\text{s.t.} \ -c_1 \leq e_j - e_\ell \leq c_1, \ j \neq \ell \qquad\qquad \leftarrow \text{ Envy-freeness constraint}$$

$$\log \Big( \frac{e_j}{1 - e_j} \Big) \cdot \tau_j \geq 0, \ j = 1, \ldots, m \qquad \leftarrow \text{ Welfare constraint}$$

$$c_2 \leq e_j \leq 1 - c_2, \ j = 1, \ldots, m, \qquad\qquad \leftarrow \text{ Feasibility constraint}$$

where $c_1 \in (0, 1)$ and $c_2 \in (0, 1/2)$. Here, the objective function captures the goal of improving information gain from study participants, which is formalized as minimizing the asymptotic variance of the inverse probability weighting estimator of the ATE (c.f. Theorem 3). The *"feasibility"* constraint restricts that the treatment assignment probability in each group is bounded away from $0$ and $1$ by a positive constant $c_1$.

To ensure fair treatment assignment and mitigate significant disparities in treatment allocations among participant groups, we introduce the *"envy-freeness"* constraint. This constraint limits the disparity in treatment assignment probabilities across different groups in an acceptable pre-specified range. The concept of "envy-freeness" originates from game theory literature and ensures that agents are content with their allocated resources without envying their peers [2, 4, 23, 41]. By incorporating this envy-freeness constraint, we address the first fairness concern and promote equitable treatment allocation.

To enhance the overall welfare of experiment participants, we introduce the *"welfare"* constraint. This constraint ensures that a group of participants is more likely to receive the treatment if their treatment effects are positive and less likely to receive the treatment otherwise. Specifically, when the group-level treatment effect $\tau_j \geq 0$, indicating that group $j$ benefits from the treatment, we want the treatment assignment probability $e_j$ to be larger than $\frac{1}{2}$. The welfare constraint achieves this by ensuring that the sign of $\log(\frac{e_j}{1-e_j})$ aligns with the sign of $\tau_j$. When incorporating the welfare constraint, we effectively address the second fairness concern by providing more treatment to beneficial groups.

## 2.3 Learning oracle strategy in adaptive experiments

In this section, we present our fair adaptive experimental design strategy for realistic scenarios where we lack prior knowledge about the underlying data distribution, and our approach achieves the same desirable properties as in the oracle setting (refer to Section 3 for justification).

We present our proposed fair adaptive experiment strategy in Algorithm 1.

---

**Algorithm 1** Fair adaptive experiment

---

**Stage 1 (Initialization):**
1: Enroll $n_1$ participants, and assign treatments in group $j$ according to $e_{1j} = \frac{1}{2}$;
2: Compute $\hat{\tau}_{1j}, \hat{\sigma}_{1j}^2(d)$, and $\hat{p}_{1j}$ as in Eq (6). Also, see the Supplementary Materials.
**Stage $t$ (Fully-adaptive experiment):**
3: **for** $t \to 2$ to $T$ **do**
4:     With $\hat{\tau}_{t-1,j}, \hat{\sigma}_{t-1,j}^2(d)$, and $\hat{p}_{t-1,j}$, solve **Problem B** to find $\hat{e}_{tj}^*$;
5:     Enroll $n_t$ participants and assign treatment with probability $\hat{e}_{tj}^*$;
6:     Update $\hat{\tau}_{tj}, \hat{\sigma}_{tj}^2(d)$, and $\hat{p}_{tj}$ as in Eq (6).
7: **end for**
**Stage $T$ (Inference):**
8: Compute $\hat{v}_j^2$ and $\hat{v}^2$ as in Eq (7).
9: Construct two-sided confidence intervals for $\hat{\tau}_j$ and $\tau$ as in Eq (8).

---

Concretely, in Stage 1 (line 1–2), because we have no prior knowledge about the unknown parameters, we obtain initial estimates of the group-level treatment effect $\hat{\tau}_{1j}$ and the associated variances $\hat{\sigma}_{1j}^2(d)$. Then, in Stage $t$ (line 4–6), our design solves the following sample analog of **Problem A** at each experimental stage:

**Problem B**

$$\min_{e} \sum_{j=1}^{m} \hat{p}_{t-1,j}\left(\frac{\hat{\sigma}_{t-1,j}^2(1)}{e_j} + \frac{\hat{\sigma}_{t-1,j}^2(0)}{1 - e_j}\right), \qquad \leftarrow \text{Minimize the estimated variance}$$

$$\text{s.t. } -c_1 \le e_j - e_\ell \le c_1, \ j \ne \ell \qquad \leftarrow \text{Envy-freeness constraint}$$

$$\log\left(\frac{e_j}{1 - e_j}\right) \cdot \hat{\tau}_{t-1,j} \ge -\delta(N_{t-1}), \ j = 1, \ldots, m \qquad \leftarrow \text{Wellfare constraint}$$

$$c_2 \le e_j \le 1 - c_2, \ j = 1, \ldots, m, \qquad \leftarrow \text{Feasibility constraint.}$$

Here, we define

$$(6) \ \hat{p}_{t-1,j} = \frac{\sum_{s=1}^{t-1} \sum_{i=1}^{n_s} \mathbb{1}_{(X_{is} \in \mathcal{S}_j)}}{\sum_{s=1}^{t-1} n_s},$$

$$\bar{Y}_{t-1,j}(1) = \frac{\sum_{s=1}^{t-1} \sum_{i=1}^{n_s} \mathbb{1}_{(X_{is} \in \mathcal{S}_j)} D_{is} Y_{is}}{\sum_{s=1}^{t-1} \sum_{i=1}^{n_s} \mathbb{1}_{(X_{is} \in \mathcal{S}_j)} D_{is}}, \quad \bar{Y}_{t-1,j}(0) = \frac{\sum_{s=1}^{t-1} \sum_{i=1}^{n_s} \mathbb{1}_{(X_{is} \in \mathcal{S}_j)}(1 - D_{is}) Y_{is}}{\sum_{s=1}^{t-1} \sum_{i=1}^{n_s} \mathbb{1}_{(X_{is} \in \mathcal{S}_j)}(1 - D_{is})},$$

$$\hat{\tau}_{t-1,j} = \bar{Y}_{t-1,j}(1) - \bar{Y}_{t-1,j}(0),$$

$$\hat{\sigma}_{t-1,j}^2(1) = \frac{\sum_{s=1}^{t-1} \sum_{i=1}^{n_s} \mathbb{1}_{(X_{is} \in \mathcal{S}_j)} D_{is}\left(Y_{is} - \bar{Y}_{t-1,j}(1)\right)^2}{\sum_{s=1}^{t-1} \sum_{i=1}^{n_s} \mathbb{1}_{(X_{is} \in \mathcal{S}_j)} D_{is}},$$

$$\hat{\sigma}_{t-1,j}^2(0) = \frac{\sum_{s=1}^{t-1} \sum_{i=1}^{n_s} \mathbb{1}_{(X_{is} \in \mathcal{S}_j)}(1 - D_{is})\left(Y_{is} - \bar{Y}_{t-1,j}(0)\right)^2}{\sum_{s=1}^{t-1} \sum_{i=1}^{n_s} \mathbb{1}_{(X_{is} \in \mathcal{S}_j)}(1 - D_{is})}.$$

One important feature of Problem B is that we introduce a relaxation of the welfare constraint through $\delta(N_{t-1})$. From a theoretical perspective, the function $\delta(\cdot)$ should be strictly positive, and satisfies $\lim_{x \to \infty} \delta(x) = 0$ and $\lim_{x \to \infty} \sqrt{x}\delta(x) = \infty$. For implementation, we recommend using $\delta(N_{t-1}) = \sqrt{\log(N_{t-1})/N_{t-1}}$. It is also possible to incorporate the standard error of the estimated

subgroup treatment effects into the welfare constraint, which motives the more sophisticated version:

$$\log\Big(\frac{e_j}{1-e_j}\Big)\cdot\frac{\hat{\tau}_{t-1,j}}{\hat{v}_{t-1,j}}\geq -\sqrt{\frac{\log(N_{t-1})}{N_{t-1}}},$$

where $\hat{v}_{t-1,j}$ is the adaptively estimated standard deviation defined in the Supplementary Materials. Scaling the welfare constraint by $\hat{v}_{t-1,j}$, a measure of randomness in $\hat{\tau}_{t-1,j}$, delivers a more clear interpretation: the above now corresponds to a t-test for the subgroup treatment effect $\tau_j$ with a diverging threshold, and the specific choice stems from Schwarz's minimum BIC rule.

After the final Stage $T$, we have the group-level treatment effect estimates $\hat{\tau}_j := \hat{\tau}_{Tj}$, the variance estimates $\hat{\sigma}_j^2(d) := \hat{\sigma}_{Tj}^2(d)$, and the group proportions $\hat{p}_j := \hat{p}_{Tj}$ (that is, we omit the time index $T$ for estimates obtained after the completion of the experiment). Together with valid standard errors, one can conduct statistical inference at some pre-specified level $\alpha$. To be precise, the estimated ATE is $\hat{\tau} = \sum_{j=1}^m \hat{p}_j\hat{\tau}_j$, and we define the following

(7)
$$\hat{v}_j^2 = \frac{1}{\hat{p}_j}\Big(\frac{\hat{\sigma}_j^2(1)}{\hat{e}_j} + \frac{\hat{\sigma}_j^2(0)}{1-\hat{e}_j}\Big), \quad\text{and}\quad \hat{v}^2 = \sum_{j=1}^m \hat{p}_j^2\hat{v}_j^2 + \sum_{j=1}^m \hat{p}_j\big(\hat{\tau}_j - \hat{\tau}\big)^2,$$

$$\text{where } \hat{e}_j = \frac{\sum_{s=1}^T \sum_{i=1}^{n_s} \mathbb{1}_{(X_{is}\in\mathcal{S}_j)} D_{is}}{\sum_{s=1}^T \sum_{i=1}^{n_s} \mathbb{1}_{(X_{is}\in\mathcal{S}_j)}}.$$

Lastly, we can construct the two-sided confidence intervals for $\hat{\tau}_j$ and $\hat{\tau}$ as

(8)
$$\Big[\hat{\tau}_j \pm \Phi^{-1}(1-\alpha/2)\cdot\hat{v}_j/\sqrt{N}\Big] \quad\text{and}\quad \Big[\hat{\tau} \pm \Phi^{-1}(1-\alpha/2)\cdot\hat{v}/\sqrt{N}\Big].$$

## 3 Theoretical investigations

In this section, we investigate the theoretical properties of our proposed fair adaptive experiment strategy, and we demonstrate that our approach achieves the same desirable properties as in the oracle setting. We work under the following assumptions:

**Assumption 1** *For $t = 1,\ldots,T$ and $i = 1,\ldots,n_t$, the covariates and the potential outcomes, $(X_{it}, Y_{it}(0), Y_{it}(1))$, are independently and identically distributed; the potential outcomes have bounded fourth moments: $\mathbb{E}[|Y_{it}(d)|^4] < \infty$ for $d = 0, 1$.*

**Assumption 2** *The group proportions $p_j$ are bounded away from 0: there exists $\delta > 0$ such that $p_j \geq \delta$ for all $j = 1, 2, \ldots, m$.*

**Assumption 3** *The sample size for each stage, $n_t$, are of the same order: there exists $c \geq 1$ such that $\frac{N}{cT} \leq n_t \leq \frac{cN}{T}$.*

Assumption 1 imposes a mild moment condition on the potential outcomes over different stages. Assumption 2 assumes that the proportion of each group is nonzero. Assumption 3 requires that the sample size in each stage are of the same order. We remark that this assumption can be easily relaxed.

**Theorem 1 (Consistent treatment effect and variance estimation)** *Assume Assumptions 1–3 hold. Then, the estimated group-level treatment effects and the associated variances are consistent:*

$$\hat{\tau}_{tj} - \tau_j = O_p\big(1/\sqrt{N_t}\big), \quad \hat{\sigma}_{tj}^2(d) - \sigma_j^2(d) = O_p\big(1/\sqrt{N_t}\big),$$

*where $N_t = \sum_{s=1}^t n_s$. As a result, after stage $T$,*

$$\hat{\tau}_j - \tau_j = O_p\big(1/\sqrt{N}\big), \quad \hat{\tau} - \tau = O_p\big(1/\sqrt{N}\big), \quad \hat{\sigma}_j^2(d) - \sigma_j^2(d) = O_p\big(1/\sqrt{N}\big).$$

Theorem 1 shows consistency of the group-level treatment effects and the associated variance estimators. This further implies the consistency of the average treatment effect estimator. The proof of Theorem 1 leverages the martingale methods [21]. Building on Theorem 1, we can establish the theoretical properties of the actual treatment allocation under our design strategy.

**Theorem 2 (Convergence of actual treatment allocation)** *Assume Assumptions 1–3 hold. Then the actual treatment allocation, defined in Eq (7), converges to the oracle allocation: $\hat{e}_j - e_j^* = o_p(1)$.*

Theorem 2 is a key result. It suggests that despite having minimum knowledge regarding the distribution of the potential outcomes at the experiment's outset, we are able to adaptively revise the treatment allocation probability using the accrued information, and the actual treatment probabilities under our proposed fair adaptive experiment strategy converge to their oracle counterparts. Building on Theorem 1 and Theorem 2, we are able to establish the asymptotic normality results of our proposed estimators and show that the standard errors are valid.

**Theorem 3 (Asymptotic normality and valid standard errors)** *Assume Assumptions 1–3 hold. Then, the estimated group-level treatment effects and the estimated ATE are asymptotically normally distributed:*

$$\sqrt{N}\big(\hat{\tau}_j - \tau_j\big) \rightsquigarrow \mathcal{N}\big(0, \, v^2(e_j^*)\big), \quad and \quad \sqrt{N}\big(\hat{\tau} - \tau\big) \rightsquigarrow \mathcal{N}\big(0, \, v^2(e^*)\big),$$

*where*

$$v_j^2(e_j^*) = \frac{1}{p_j}\Big(\frac{\sigma_j^2(1)}{e_j^*} + \frac{\sigma_j^2(0)}{1 - e_j^*}\Big), \quad and \quad v^2(e^*) = \sum_{j=1}^{m} p_j^2 v_j^2(e_j^*) + \sum_{j=1}^{m} p_j(\tau_j - \tau)^2.$$

*In addition, the standard errors in Eq (7) are consistent: $\hat{v}_j^2 - v_j^2(e_j^*) = o_p(1)$ and $\hat{v}^2 - v^2(e^*) = o_p(1)$.*

Theorem 3 shows the asymptotic normality results of the estimated treatment effects under our proposed adaptive experiment strategy. In addition, Theorem 3 verifies that the constructed confidence intervals in Eq (8) attain the nominal coverage thanks to the consistency of standard errors. The proof of Theorem 3 relies on the convergence of the actual treatment allocation in Theorem 2 and and the martingale central limit theorem [21].

## 4  Simulation evidence

In this section, we evaluate the performance of our proposed fair adaptive experiment strategy through simulation studies. We summarize the takeaways from the simulation studies as follows. First, our proposed fair adaptive experiment strategy achieves higher estimation efficiency than the complete randomization design. Second, compared to a classical adaptive experiment strategy, our method avoids disproportionate treatment assignment probabilities across different groups of participants and accounts for participants' welfare.

Our simulation design generates the potential outcomes under two data-generating processes. **DGP 1**: Continuous potential outcomes $Y_i(d)|X_i \in \mathcal{S}_j \sim \mathcal{N}(\mu_{d,j}, \, \sigma_{d,j})$, where $\boldsymbol{\mu}_1 = (1, 4)^\mathsf{T}$, $\boldsymbol{\mu}_0 = (4, 2)^\mathsf{T}$, $\boldsymbol{\sigma}_1 = (2.5, 1.2)^\mathsf{T}$, and $\boldsymbol{\sigma}_0 = (1.5, 3.5)^\mathsf{T}$. The group proportions are $\boldsymbol{p} = (0.5, 0.5)^\mathsf{T}$. The group-level treatment effects are $\boldsymbol{\tau} = (-3, 2)^\mathsf{T}$. **DGP 2**: Binary potential outcome: $Y_i(d)|X_i \in \mathcal{S}_j \sim$ Bernoulli$(\mu_{d,j})$, where $\boldsymbol{\mu}_1 = (0.6, 0.2, 0.3, 0.4, 0.1)^\mathsf{T}$, $\boldsymbol{\mu}_0 = (0.1, 0.5, 0.3, 0.4, 0.6)^\mathsf{T}$. The group proportions are $\boldsymbol{p} = (0.15, 0.25, 0.2, 0.25, 0.15)^\mathsf{T}$. To mimic our first case study (in Supplementary Materials), consider the log relative risk as the parameter of interest: $\log \mathbb{E}[Y(1)] - \log \mathbb{E}[Y(0)]$. The group-level treatment effects are $\boldsymbol{\tau} = (1.79, -0.92, 0, 0, -1.79)^\mathsf{T}$.

We compare three experiment strategies for treatment assignment: (1) our proposed fair adaptive experiment strategy, (2) the doubly adaptive biased coin design (DBCD) [70], and (3) the complete randomization design, which fixes the treatment allocation probability to be $1/2$ throughout the experiments. To mimic the fully adaptive experiments, we fix stage 1 sample size at $n_1 = 40$ and $n_t = 1$ for $t = 2, \ldots, T$, where the total number of stages ranges from $T \in \{40, \ldots, 400\}$. We evaluate the performance of each strategy from two angles. First, we compare the standard deviation of the ATE estimates to evaluate the estimation efficiency. Second, we compare the fraction of participants assigned to the treatment arm in each group to evaluate the fairness in treatment allocation. The simulation results are summarized in Figure 1.

We first focus on (A) and (B), which correspond to DGP 1. Panel (A) depicts standard deviations of the treatment effect estimates under the three experiment designs. It clearly demonstrates that our proposed method achieves higher estimation efficiency compared to complete randomization.

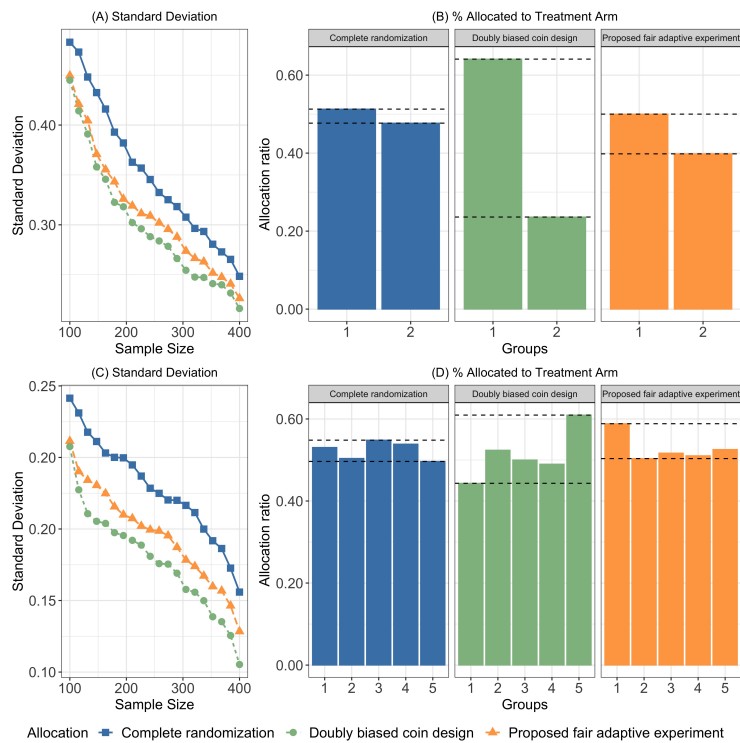

Figure 1: Comparison of the proposed adaptive experiment design strategy, the complete randomization design, and the doubly adaptive biased coin design. (A) and (C) show the standard deviation comparisons. (B) and (D) show the percentage of participants allocated to the treatment arm in each group under different experiment strategies.

Panel (B) shows the treatment assignment probabilities produced by the three experiment design strategies. Not surprisingly, complete randomization allocates 50% participants to the treatment arm regardless of their group status. On the other hand, the DBCD design may produce extreme treatment allocations. In addition, participants in different groups may receive drastically different treatment allocations, which can raise fairness concerns. Encouragingly, our approach generates treatment assignment probabilities that not only are closer to 50% (i.e., less extreme) but also exhibit less variation across groups. Panel (C) and (D) summarize the simulation evidence for DGP 2 in which the outcome variable is binary. A similar pattern emerges: our fair adaptive experiment design approach improves upon complete randomization, delivering more precise treatment effect estimates. It also accounts for fairness and participants' welfare in assigning treatments.

The simulation results demonstrate the clear trade-off between fairness/welfare and statistical efficiency by adopting our proposed fair adaptive experiment strategy. Although it involves a minor sacrifice in estimation efficiency when contrasted with the DBCD design, our approach delivers more fair treatment allocations and safeguards participant well-being. As our proposed method does not restrict each group to have exactly the same treatment assignment probabilities as in the complete randomization design, it improves the estimation efficiency of ATE. We provide additional simulation results and synthetic data analyses in the Supplementary Materials.

## 5   Discussion

In this work, we propose an adaptive experimental design framework to simultaneously improve statistical estimation efficiency and fairness in treatment allocation, while also safeguarding participants' welfare. One practical limitation of the proposed design is that its objective mainly aligns with the experimenter's interests in estimating the effect of a treatment, as opposed to the interests of the enrolled participants. This aspect offers opportunities for future research exploration.

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
