# OpenReview forum: "Fair Adaptive Experiments"
_NeurIPS.cc/2023/Conference — NeurIPS 2023 poster_

### Official Review · Reviewer_gk4J · 2023-06-20

**Soundness:** 3 good
**Presentation:** 3 good
**Contribution:** 3 good
**Rating:** 6
**Confidence:** 4

**Summary:**

This paper investigates adaptive experimental design with a focus on fairness. In this context, fairness refers to ensuring that the probabilities of treatment assignment do not significantly differ across various groups. The authors propose a fair adaptive experimental design that simultaneously enhances data use efficiency, achieves an “envy-free” treatment assignment guarantee, and improves the overall welfare of participants.

**Strengths:**

1.	Fairness is gaining increasing importance in various fields, including experimental design.
2.	The technical parts (Section 2 and Section 3) are solid, well organized and clearly presented.
3.	Both simulation studies and a case study based on real data are provided.




**Weaknesses:**

1.	Regarding the definition of 'fairness,' this paper considers fairness as the requirement for treatment adoption probabilities to be similar across different groups. However, in many scenarios, particularly in clinical trials, such a definition of fairness may not be entirely convincing. The reason why we want to consider the covariate $X_{it}$ and group $\mathcal{S}_j$ is because the treatment effect may be very different among different groups. For instance, in the case of patients grouped by biomarkers into $\mathcal{S}_1$ and $\mathcal{S}_2$, the new treatment yields a highly positive effect in $\mathcal{S}_1$ but a strongly negative effect in $\mathcal{S}_2$. Consequently, enforcing close treatment probabilities in these two groups might be deemed unfair. It would be beneficial if the authors could provide motivating examples in the introduction that align well with the current formulation.
2.	Regarding the consideration of 'welfare,' I have some uncertainty about its direct relevance to the main topic of 'fairness.' While I understand that enforcing fairness constraints may potentially impact welfare, I'm unsure about the specific implications if we were to remove the welfare constraint in Problem A. In lines 66-70, the authors briefly touch upon this question, but the points made are not entirely clear to me. It would be beneficial if the authors could further elaborate on why “the second fairness concern arises when the adaptive treatment allocation does not adequately account for the overall welfare of experimental participants.”
3.	About the feasibility of the optimization Problem A and Problem B. In my view, it seems that Problem A could be infeasible, especially when $c_1$ is very small or $c_2$ is very close to ½. I am curious about whether the authors once met the infeasibility issue in the simulation studies and the case study.
4.  Regarding the length of the first stage, it would be beneficial to include additional comments in the paper on how to select the value of $n_1$. Furthermore, I would appreciate a clearer explanation of how exactly $n_1$ influences the main results, as this aspect is important for my understanding. Additionally, in lines 107-108, the authors mention that 'An important methodological and practical innovation of our framework is that it does not require the number of participants enrolled in the first stage to be proportional to the overall sample size.' However, more supporting comments on this claim within the main text are expected. Providing some additional explanation or evidence for this statement would strengthen the paper's argument.
5. Regarding the objectives of the experiment, in lines 76-78, the authors state that 'we propose a fair adaptive experimental design strategy that balances competing objectives: improving fairness, enhancing overall welfare, and gaining efficiency. Our strategy strikes a delicate balance between these trade-offs.' However, it remains unclear what trade-off means among these three objectives. While I can comprehend the trade-off between enhancing overall welfare and gaining efficiency, as it has been previously investigated in [1][2]. I find it unlikely that improving fairness could simultaneously trade off with two objectives that already have a trade-off between them.  I believe it would be highly beneficial if the authors could provide further insights into the nature and rationale behind the trade-off that exists among these objectives. Expanding on this aspect would provide a deeper understanding of the decision-making process and the underlying considerations in the proposed strategy.

Reference:

[1] Erraqabi, A., Lazaric, A., Valko, M., Brunskill, E., & Liu, Y. E. (2017, April). Trading off rewards and errors in multi-armed bandits. In Artificial Intelligence and Statistics (pp. 709-717). PMLR.

[2] Simchi-Levi, D., & Wang, C. (2023, April). Multi-armed bandit experimental design: Online decision-making and adaptive inference. In International Conference on Artificial Intelligence and Statistics (pp. 3086-3097). PMLR.




**Questions:**

Apart from the major points above, I have several minor points.

1)	The paper is well organized in general. However, I think the writing could be further improved. For the technical parts (Sections 2 and 3), after each lemmas and theorems, it will be better to provide more insights and discussions.

2)	In Lemma 1, it is more appropriate to denote the oracle treatment assignment probablity for group j under the proposed welfare constraint as $e^∗ _{j,proposed}$ instead of $e^∗ _{j,alternative}$.

3)	For a conference paper, I believe the abstract could be more concise as it is currently a bit long.


**Limitations:**

See previous comments.

---

> ### Author Rebuttal · Authors · 2023-08-09
>
> Thank you for your insightful suggestions and questions!
> - Thank you for inspiring us to delve deeper into the concept of fairness. In clinical trials, only when a drug demonstrates no adverse effects in Phase I trials does it advance to Phase 2 & 3 trials, where its effectiveness is rigorously evaluated. Thus, by the time a drug reaches Phase 2 and 3 (which are our design focus), the likelihood of it causing harm to patients is significantly diminished. In this case, as the drug is unlikely to negatively impact patients, our design aims to ensure experimental fairness by assigning distinct individuals a roughly equal opportunity to receive the medication. Furthermore, in our current case study, the treatment is a strategy that bundles health insurance with microfinance, and subgroups are defined by household income levels. Ensuring fair treatment allocation is important because it allows subgroups with different financial statuses to have similar exposure to treatment.
> - Regarding the consideration of welfare, if the welfare constraint is removed from Problem A, the derived treatment allocation strategy will only depend on subgroup variances and the difference between subgroup treatment allocations. In this case, subgroup treatment effects do not play a role in the decision-making process, and the overall welfare cannot be enhanced. In fact, our motivation for adding the welfare constraint aligns with your insightful point: we hope the participants' differential responses to the treatment could also play a role in the decision-making process. The welfare constraint says that if the subgroup treatment effect is positive, meaning the treatment is beneficial, our design tends to assign more subjects to the treatment arm in this subgroup.
>    - We provide insights regarding the effect of welfare constraint on treatment allocation in Figure 3 in the manuscript. Figure 3 (B) shows the treatment allocation under fairness and welfare constraint; Figure 3 (C) shows the treatment allocation only under fairness constraint. We observe that the treatment assignment probability for Group 1 is reduced when adding the welfare constraint because Group 1 negatively reacts to the treatment. Thus, the welfare constraint acts as additional protection against potentially harmful treatment in certain subgroups.
> - We are sorry if the presentation of Problem A (and B) has caused confusion. Both problems are indeed feasible, as the constraints always lead to a nonempty feasible region. (i) Take Problem A as an example. We let $\delta = \min${ $0.5c_1, 0.5-c_2$ }$> 0$, then the set of candidate assignment rules are $(e_1,e_2,\dots,e_m)\in [0.5-\delta ,0.5+\delta]^{m}$. In other words, it is always feasible to adopt an assignment probability close to $1/2$. If one sets $c_1$ to a small number and $c_2$ close to $1/2$, then the algorithm will return an assignment rule resembling complete randomization. (ii) In our numerical investigation, while we never observe the constraints leading to an empty feasible region, we do see that the constraints can be binding -- especially the fairness constraint. For example, the middle panel in Figure 2(B) for DBCD design: without incorporating fairness concerns, the assignment probabilities can vary substantially across groups and take extreme values. On the other hand, the right panels in 2(B) and 2(C) correspond to our method, where the treatment probabilities are much less extreme. This means that the fairness constraints can be binding in this synthetic data-based numerical experiment.
> - We realize that the assumption on $n_1\rightarrow \infty$ can be dropped (please find the new proof in our response to Reviewer 3). In practical implementation, following existing adaptive experiment literature (e.g., Asymptotic properties of doubly adaptive biased coin designs for multitreatment clinical trials), we suggest that in each subgroup, assigning at least $2$ subjects to each of the treatment arms.
> - We greatly appreciate you pointing out Simchi-Levi and Wang (2023), which is very helpful to us.
>   - We find the minimax lower bound and the Pareto frontier characterized by Simchi-Levi and Wang (2023) very enlightening. Indeed, using their notation, our procedure is Pareto optimal: the treatment effect estimator we propose has error $\tilde{O}(1/\sqrt{N})$. Since we require the treatment assignment probability to be bounded away from zero, the regret in our case is linear in the sample size. These two observations suggest that our procedure achieves the minimax lower bound of Simchi-Levi and Wang (2023).
>   - We believe our current approach potentially complements Simchi-Levi and Wang (2023) from two perspectives. On the one hand, Simchi-Levi and Wang (2023) focus on the trade-off between regret and statistical estimation efficiency. The goal of our paper, however, is to provide an adaptive procedure for efficient treatment effect estimation while also taking fairness and welfare into consideration. We thus believe that our procedure is useful for experimental settings where uncovering the underlying causal effect/mechanism is of central importance. On the other hand, since our paper mainly focuses on the efficient estimation of treatment effects under fairness and welfare constraints, we provide statistical guarantees beyond the rate of convergence. Specifically, we characterize the asymptotic normality of the proposed estimator and its variance estimator, which together enable valid statistical inference. We plan to add a comprehensive review to our revision about Simchi-Levi and Wang (2023).
>
> Thank you for your helpful suggestions on improving the writing. We plan to make the following revisions:
> - Add more discussions, insights into the theoretical results, related references, and practical guidance in Sections 1-3.
> - Revise the notation for the oracle treatment assignment probability as $e^*_{j,\text{proposed}}$.
> - Shorten the abstract to make it more concise.

---

> > ### Comment · Reviewer_gk4J · 2023-08-10
> >
> > Thank you for your efforts in addressing my comments. From my standpoint, all of my concerns have been well resolved.

---

> > > ### Author Response · Authors · 2023-08-11
> > >
> > > Thank you for your reply and recognition. We appreciate your time and valuable feedback.

---

### Official Review · Reviewer_62S1 · 2023-06-28

**Soundness:** 2 fair
**Presentation:** 2 fair
**Contribution:** 1 poor
**Rating:** 2
**Confidence:** 3

**Summary:**

The authors delve into an examination of adaptive experimental design subject to the constraints of fairness. They portray a scenario where a researcher, within a sequential experimental context, adapts the allocation between control and treatment groups among a cohort of subjects to optimize the statistical efficiency of the average treatment effect estimation. At the same time, the researcher strives to maintain a balanced proportion of treatments across various demographic groups and maximize the total welfare of these treatments, a condition set forth to uphold fairness. The main contribution of this manuscript lies in the formulation of a novel strategy for treatment allocation that simultaneously adheres to the constraints of fairness. The authors provide a theoretical analysis of their devised method, validating that the average treatment effect estimated from their approach exhibits asymptotic consistency and adheres to a central limit theorem-like convergence toward a particular Gaussian distribution. Empirical studies, leveraging synthetic simulations, demonstrate their method's competitive statistical efficiency against existing approaches while ensuring fairness is upheld. Furthermore, the efficacy of the developed method is also confirmed via empirical simulations that mimic real data, further solidifying the reliability and practicality of their method.

**Strengths:**

This paper investigates the fairness problem in the adaptive experimental design. This problem is well-motivated and of particular interest to NeurIPS's community. The experimental results adequately demonstrate the proposed method's competitive efficiency and guarantee of fairness.

**Weaknesses:**

This paper lacks consideration of essential existing contributions, specifically those of
- V. Hadad et al. Confidence Intervals for Policy Evaluation in Adaptive Experiments. Proceedings of the National Academy of Sciences, 2021.
- D. Simchi-Levi et al. Multi-armed Bandit Experimental Design: Online Decision-making and Adaptive Inference. AISTATS, 2023.
- M. Kato et al. Adaptive Experimental Design for Efficient Treatment Effect Estimation. NeurIPS 2020 Workshop on Causal Discovery & Causality-Inspired Machine Learning.

Notably, the findings put forth by Hadad et al. warrant particular attention, given their methodological relevance to this research. Hadad et al. introduce an off-policy estimation of the average treatment effect, a method that works regardless of the treatment assignment mechanism. They also validated the asymptotic consistency and normality of their approach. While Hadad et al.'s method operates without covariates, unlike the scenario in the present paper, this discrepancy does not diminish the applicability of their method. This paper's research problem can be recast into a non-covariate one by transforming the experimental process into group-wise processes. Furthermore, with an appropriately selected set of parameters, the method discussed by Hadad et al. can be reduced to this paper's proposed solution. Consequently, the asymptotic consistency and normality results offered by Hadad et al. would translate to the circumstances of this paper, which seamlessly results in Theorems 1 and 2.

Additional insightful points can be gleaned from the work of Kato et al., who propose a treatment assignment mechanism to minimize the variance of the average treatment effect. While unconstrained, their method presents intriguing similarities with the constrained approach of the current paper. Appendix F of Kato et al., particularly noteworthy, introduces an ethical consideration in the treatment assignment process by incorporating a fairness constraint.

The omission of these existing methods from the current discussion and the direct applicability of these approaches to the circumstances of this paper leads me to question the originality of the proposed research. The paper would benefit from a comparative evaluation with these significant contributions to ensure that it advances the related field.


The present paper is unfortunately beset by a number of quality issues that necessitate substantial revisions. To begin with, the paper neglects to outline the assumptions required for causal inference. Particularly, the authors' methodology employs the inverse propensity score weighting technique. This approach necessitates the fulfillment of certain assumptions, encompassing consistency, no unmeasured confounders, and positivity. The lack of explaining these assumptions calls into question the validity of the resultant findings and negatively impacts the overall quality of the paper.

Moving to Section 2.2, it is observed that quality concerns exist. Specifically, Lemma 1's statement introduces the concept of an alternative constraint without providing its rigorous definition. This omission makes us hard to ascertain the correctness of the lemma. Additionally, the section contains considerable redundancy, particularly evident between the contents of Lines 209-219 and Lines 220-230. Such repetition impairs the overall readability of the paper and dilutes the impact of the information conveyed. Cumulatively, these quality issues have contributed to a lower assessment of the paper's presentation score in this review.

**Questions:**

I suggest that the authors elucidate a specific scenario wherein fairness issues arise and illustrate how the applied fairness constraint alleviates these problems. The fairness constraints utilized by the authors appear to diverge from conventional ones; thus, it is essential for the authors to rationalize their choice to implement the proposed fairness constraint. Illustrating a specific scenario will be instrumental in justifying the use and effectiveness of the fairness constraint.

The authors state that their formulated fairness constraints are based on the principles of envy-freeness and total welfare. Nevertheless, there may be a potential misapprehension of these principles by the authors as their derived fairness constraints deviate from the standard definitions established from the above notions. Both principles are rooted in the game theory literature and are presented in a context where each game participant possesses a distinct utility function. This utility function is exploited to evaluate the benefits derived from a specific treatment. Envy-freeness and total welfare can be viewed as inherent characteristics of the participants' utility contingent upon their treatments. In essence, the constructs of envy-freeness and total welfare are fundamentally anchored on utility functions. Consequently, it is recommended that the authors shed light on the nature of the utility functions leveraged in their investigation. By doing so, they ensure a more precise understanding of the fairness constraints.

In this paper, the authors endeavor to validate Theorems 1 and 2 by invoking Lemma A.1. However, the assertion of Lemma A.1 appears to be flawed, particularly in the analysis of the conditional expectation of $Z_t$ given $\mathcal{F}{t-1}$. This is because $E[Z_t|\mathcal{F}{t-1}]$ is not a random variable and hence cannot equate to $Z_{t-1}$, evident by $E[D_tY_t1(X_t \in \mathcal{S}j)|\mathcal{F}{t-1}] = p_j\hat{e}^*{t,j}E[Y_t|D_t=1,X_t \in \mathcal{S}j]$. However, despite this ostensible flaw in Lemma A.1, it is plausible that the statements of Theorems 1 and 2 might be correct. The validity of these theorems may be independently corroborated by employing the known fact that $Z_t$ possesses a common mean. This subsequently implies that the process of $Z_t$ minus the common mean satisfies the definition of a martingale difference sequence.

**Limitations:**

There are no specific limitations or potential negative social impacts that require discussion in relation to the current method.

---

> ### Author Rebuttal · Authors · 2023-08-09
>
> Thank you for your feedback and pointing out these three references.
> - V. Hadad et al. discuss off-line policy evaluation, where experimental data have been collected. Their work focuses on analyzing experimental data to estimate ATE rather than designing randomized experiments (which is the focus of our work). In particular, their adaptively weighted AIPW estimator aims to handle the challenging scenario where the propensity score tends to zero (i.e., violation of positivity assumption). In contrast, as the positivity assumption holds by design in our setting (the feasibility constraint in Problem A), our estimator and the proof technique are both different from V. Hadad et al.
> - We provided a comparison with D. Simchi-Levi and Wang (2023) in the response to the 4th reviewer.
> - Our work differs from M. Kato et al. from at least four perspectives (i)The treatment effect estimator in M. Kato et al. requires estimating outcome models, which is challenging in the presence of correlated data in response-adaptive experiments. However, our IPW estimator(with estimated propensity scores) is simpler to implement and attains the same efficiency bound. Though M. Kato et al. also mentioned the IPW approach(with known propensity scores), it is unclear whether their IPW estimator can attain the semiparametric efficiency bound. (c.f. Hirano, Imbens, and Ridder, 2003 Econometrica: "Efficient estimation of average treatment effects using the estimated propensity score"). (ii)Their Assumption 1 imposes a boundedness assumption on the treatment assignment probability,  while in our design, the derived treatment assignment probabilities are naturally bounded due to our feasibility constraint. (iii)Their variance estimator and the confidence intervals are not constructed. Thus, it is not possible to construct a pointwise confidence interval for the average treatment effect. (iv)Appendix F in  M. Kato et al. contains two fair design directions, which differ greatly from our proposal. Their first design is to bound the overall treatment assignment probability by a constant. The second design is to have Pareto optimal treatment allocation.  Compared to these directions, our design centers around "group" fairness, aiming to enhance participants' well-being while avoiding extra fairness complications among distinct individuals.
> - Regarding assumptions required for causal inference, we do not need to impose the causal assumptions mentioned in your review (consistency, no unmeasured confounders, and positivity), because these assumptions automatically hold in designing randomized experiment settings.  For example, the positivity assumption is satisfied by our feasibility constraint in Problem A: $c_2\leq e_j\leq 1-c_2$, where $c_2\in (0,1/2)$ is a constant.
> - Our fairness constraints are indeed rooted in game theory.  In our context, the "utility function" is essentially the subgroup treatment assignment probability $e_j$. Our primary motivation for having the envy-freeness constraint is to ensure a more balanced treatment allocation across subgroups. Yet, this constraint does not consider the possible heterogeneous effects of subgroups. To address this, we add welfare constraint. While welfare has been discussed in many fairness literature, its interpretation depends on the context. Our context is to design adaptive experiments that assign treatment assignments fairly yet skew more towards benefiting subgroups. To meet this design objective, our welfare constraint takes a specific form as shown in the manuscript. For example, in our case study, we hope subgroups that positively react to the financial strategy can have a higher treatment assignment.
> -  The definition of the alternative welfare constraint mentioned in Lemma 1 was introduced in lines 229 to 230 above Lemma 1 in our current manuscript.
> - We apologize that the construction of $Z_t$ in the original supplement was not clear. After inspecting our proof, we find a neater way of proving Theorem 1 (without using Lemma A.1 at all): As we are in a fully-sequential setting, both $T$ and $N$ go to infinity. We will use $T$ and $N$ interchangeably in the following proof. Without loss of generality, we set $n_t=1$. Theorem 1 proof relies on Assumptions 1-2. For each time $t$, we define the filtration $\mathcal{F}_t$ as the $\sigma$-algbra formed by {$Y_1,\ldots,Y\_{t-1}$} and {$X_1,\dots,X\_{t-1}$}. For arm $d\in${$0,1$} and subgroup $j\in\{1, \ldots,m\}$, we define a sequence of stopping times: $\tau\_{k,d,j} =$ min{ $t: N\_{d,j}(t) = k$}, where $N\_{d,j}(t)=\sum\_{s=1}^t1(D_s = d) 1(X_s\in\mathcal{S}_j)$. Our construction says that given $\tilde{\mathcal{F}}\_{t-1}$, the probability of the joint event that $1(D_t = d)1(X\_t\in\mathcal{S}\_j)$ is strictly bounded away from 0 by a constant,  and therefore Theorem 4.3.4 of Durrett (2019) implies that $P[\limsup\_{t\to\infty}1(D\_t = d)1(X\_t\in\mathcal{S}\_j) = 1] = P[\lim\_{T\to\infty}N\_{d,j}(T) = \infty] = 1.$ In other words, the stopping time, $\tau\_{k,d,j}$, defined above is finite for all $k$ almost surely. Now take, for example, $d=1$. Following Doob (1936), $\{Y\_{ \tau\_{k,1,j}}:\ k=1,2,\dots\}$ is an iid sequence with the distribution $Y(1)|X\in \mathcal{S}\_j$. To establish the consistency result, we have the following equations held by definition: $\frac{ \sum\_{s=1}^{T} 1\_{(X\_{s}\in\mathcal{S}\_j)}D\_{s}Y\_{s}}{\sum\_{s=1}^{T} 1\_{(X\_{s}\in\mathcal{S}\_j)}D\_{s}} =  \frac{\sum\_{k=1}^{N\_{1,j}(T)}  Y\_{ \tau\_{k,1,j}}(1)}{N\_{1,j}(T)}.$
>
> Per your suggestions, we plan to make the following revisions
>  - Add literature reviews on the three references;
> - Update Theorem 1 proof and remove Lemma A.1;
> - Provide more discussion on the fairness constraints making connections to the game theory literature;
> - Remove the redundancy in lines 209 - 230.
> Reference:
> 1. Doob, J. L. (1936). Note on probability. Annals of Mathematics.
> 2. Durrett, R. (2019). Probability: theory and examples, volume 49.

---

> > ### Comment · Reviewer_62S1 · 2023-08-13
> >
> > Thank you for your detailed rebuttal. After thoroughly reviewing the points made by the authors, my perspective remains largely consistent with my initial review. When comparing this paper to the ones I initially cited, I still find the paper's contribution to be somewhat limited. A more detailed response is provided below.
> >
> > > Contribution compared to existing research
> >
> > I appreciate the authors' emphasis on how this paper distinguishes itself from much of the existing work, specifically by focusing on the design of randomized experiments rather than merely analyzing experimental data to estimate ATE. To my understanding, the central results of this paper hinge on the consistency of the proposed randomized experimental design. The papers I cited earlier also discuss the consistency of arbitrary randomized experimental designs under specific assumptions, which seem to align with the proposed design. The ATE estimation method introduced resembles an AIPW estimator with appropriate hyper-parameters, including weight choices. Therefore, in terms of consistency, the paper seems to be in sync with the analytical scope present in existing research, leading me to view its contribution as relatively modest.
> >
> > While I acknowledge the novelty of the proposed randomized experimental design, its unique aspects appear to be centered around the introduction of specific constraints. Its consistency is recognized, but it largely aligns with existing analyses. The broader utilities of the proposed design seem less explored. For example, the proposed method strives for treatment allocation that adheres to fairness constraints, such as envy-freeness and welfare constraints. However, the resultant optimization problem does not appear to present notable technical challenges. Hence, I perceive this paper as primarily introducing a novel fairness constraint to the randomized experimental design without thoroughly evaluating its benefits beyond consistency. I encourage the authors to delve deeper into the challenges of developing the proposed randomized experimental design.
> >
> > > Assumptions
> >
> > I acknowledge that the proposed randomized experimental design does not depend on the positivity assumption. Yet, there seems to be no fundamental difference between employing a positivity assumption and introducing a feasibility constraint. While the non-requirement of the positivity assumption is highlighted, its significance appears less pronounced.
> >
> > I am surprised that an assumption regarding unmeasured confounders is deemed unnecessary. I recommend that the authors provide a rigorous proof to substantiate this claim.
> >
> > > Envy-freeness
> >
> > From what I understand, envy-freeness does not inherently guarantee a balanced treatment allocation among subgroups. Envy-freeness suggests that an individual's utility exceeds what they might have received from another treatment rather than ensuring a balanced treatment. I'd like to point out that the notion of fairness presented may deviate from a traditional interpretation of envy-freeness.

---

> > > ### Author Response · Authors · 2023-08-13
> > >
> > > We are sorry to hear that you still have concerns about our paper.
> > >
> > > - You repeatedly mentioned the phrase “consistency of the proposed randomized experimental design,” but it is not a standard terminology in the experimental design literature. As such, we could not understand your comment.
> > >
> > > - You also suggested that “the resultant optimization problem does not appear to present notable technical challenges.” However, we would like to clarify that we are not trying to deliberately construct a complex optimization algorithm, as this is not the goal of our paper. The goal of our paper is to propose an adaptive experimental design method incorporating fairness and welfare concerns, with solid statistical guarantee (such as asymptotic normality and efficiency). In fact, as randomized field experiments are usually very costly, practitioners tend to avoid procedures that are too complex. For this reason, we believe it is a advantage that our procedure does not involve complicated numerical optimization, a feature that makes our proposal more transparent.
> > >
> > > - Regarding the “no unmeasured confounders” assumption, we have explained that this assumption holds by construction. In randomized experiments, the treatment assignment is (conditionally) independent of the potential outcomes by construction, and therefore the assumption holds trivially. This is why we do not need to explicitly impose this assumption, and no proof is needed. We note that this is not unique to our paper. For example, Hu and Zhang (2004) is a classical reference. You and another reviewer mentioned the recent work of Simchi-Levi and Wang (2023). None of these two papers had to impose the "no unmeasured confounders" assumption.

---

### Official Review · Reviewer_fjNx · 2023-07-07

**Soundness:** 3 good
**Presentation:** 3 good
**Contribution:** 3 good
**Rating:** 6
**Confidence:** 4

**Summary:**

This paper is concerned with the problem of experimental design for adaptive experiments under a fairness constraint, i.e., the restriction that some individuals not be disproportionately more likely to receive treatment. The authors approach this problem by modifying the usual Neyman allocation with constraints for envy, welfare, and feasibility of the problem. Update rules are provided for the problem to provide treatment probabilities at each round. The authors also provide theoretical results which bound the ATE and variance, and convergence to the oracle strategy in the limit.

**Strengths:**

* This is a novel task formulation for an important problem. The authors do a nice job of motivating the problem as well, and provide a clear description of the desired characteristics of the solution.

* The proposed algorithm is simple and efficient. The authors do a nice job of laying out both problems A and B and clearly describing the update procedure.

* Provided theory is nice in being able to reason about the behavior of the proposed procedure. The bound on the ATE and variance is very reasonable, with the proposed procedure taking only a modest penalty in terms of convergence rate. Asymptotic normality and convergence to the optimal strategy in the limit are also nice results.



**Weaknesses:**

While the strategy converges to the optimal strategy in the limit, it's not clear to me how much of an effect these parameters have on the procedure's behavior in finite samples. This is especially important in the design setting, where the main motivation of adaptivity is improving finite sample performance. It would be nice to see some simulated/empirical evidence that examines the robustness of the proposed procedure to misspecification. There is also a natural tension between the welfare and fairness constraints, it would be helpful if the authors provide a discussion of this. It also would have been useful if the authors provided a comparison to a non-adaptive (but balanced) procedure a as an additional baseline (e.g., rerandomizatoin, Gram-Schmidt walk). In this setting, treatment probabilities would remain fixed but variance would still be minimized by accounting for group status with an obvious tradeoff on the welfare constraint.


**Questions:**

* How does this procedure compare with adding regularization to a simple adaptive procedure like UCB?
* Little guidance is given for selecting the constraints. In practice how much of an effect do these have on relative performance? Can the authors provide a sense of worst case behavior here?

**Limitations:**

The authors overall do a nice job of motivating the problem of fairness in treatment allocation. However, I do think that the introduction of some of these constraints, while well motivated, still leaves a subjective decision in the hands of the experimenter. It seems given this that more attention/discussion should be given to this choice in light of the example used in the paper which has a significant impact on the lives of people who are subjected to the experiment.

---

> ### Author Rebuttal · Authors · 2023-08-08
>
> We greatly appreciate your insightful feedback!
>
> - Regarding the finite sample performance of our design, we provide the following figures:
>   - In the attached pdf file, Figure 1 shows the empirical treatment allocation for Group 1 following DGP1 in our manuscript. The empirical treatment allocation refers to the treatment assignment probability derived from solving optimization Problem A in our manuscript. The dashed line indicates the oracle treatment allocation. Figure 1 shows that the empirical subgroup treatment allocation converges to the oracle treatment allocation even when the sample size is finite.
>   - In Figures 2 (A) and (C) in the manuscript, we show the estimation efficiency with respect to various sample sizes under our design compared to other state-of-the-art designs. Figures 2 (A) and (C) in the manuscript demonstrate that our design has a rather high estimation efficiency in finite samples.
> - There is indeed a natural tension between the fairness and welfare constraints. In our manuscript, Figure 3 (B) shows the subgroup treatment allocations under the welfare constraint, while Figure 3 (C) shows the scenario without it. In the absence of the welfare constraint but with the fairness constraint in place, as in Figure 3 (C), Group 1 receives the highest treatment assignment probability among all subgroups. However, when both fairness and welfare constraints are present, as seen in Figure 3 (B), the treatment assignment probability for Group 1 is  reduced under our design. This is because Group 1 shows negative response to the treatment. Intuitively, to protect subjects in Group 1 from being harmed by the treatment, our design reduces the treatment assignment probability in Group 1 when the welfare constraint is present.
> - Thank you for mentioning the comparison between our design and rerandomization. We provide a comparison of our proposed design with rerandomization in Figure 2 in the attached pdf file.
>    - The simulation setup follows DGP 1 in our manuscript, and we further generate an additional covariate $W_{i}\sim \text{Bernoulli}(0.4)$, $i=1,\ldots, N$. Given that rerandomization requires a pre-specified treatment assignment probability, we use the oracle subgroup treatment assignment probabilities derived from our design. Then, within each subgroup, we perform rerandomization to balance the covariate $W$.
>    - Figure 2 in the attached pdf file shows that, given known oracle treatment assignment probabilities, the empirical treatment allocation using rerandomization can approximate the oracle allocation quite closely. However, when the oracle treatment assignment probabilities are unknown, our design can sequentially approach the oracle treatment allocation by learning from the collected data.
> - The UCB procedure indeed shares some similarities with our procedure, as they both adaptively revise experiment strategies to achieve experimental objectives with a higher data use efficiency. In addition, adding regularization to the UCB procedure is also a very nice approach to incorporate fairness constraints. However, there are several differences between our procedure and the UCB procedure. The design objective of the UCB procedure is to identify the best arm by maximizing rewards, while our design objective is to minimize the variance of the estimated average treatment effect. The UCB procedure revises the sampling proportion of each arm while we revise the treatment assignment probabilities within each subgroup. The sampling proportion of each subgroup remains fixed throughout our experiment.
> - As for the selection of fairness parameter, we provide some guidance on selecting the fair constraints in Figure 3 in the attached pdf file.
>    -  Figure 3 in the attached pdf file compares estimation efficiency under different values of $c_1$. $c_1$ appears in our envy-freeness constraint, which is defined as $-c_1\leq e_i -e_j \leq c_1$, $i\neq j$, $c_1 \in (0,1)$. We observe that the lowest estimation efficiency corresponds to the smallest $c_1$, suggesting that the fairest treatment allocation might lead to a compromise in estimation efficiency.
>     - We recognize that the "optimal" balance between estimation efficiency and fairness can vary based on the specific context of application. For a general guidance, we recommend setting $0.1\leq c_1\leq 0.3$ when fairness concern is slightly prioritized over estimation efficiency. Conversely, when estimation efficiency is slightly prioritized over fairness, we recommend setting $0.3\leq c_1\leq 0.5$ to achieve a more balanced trade-off between estimation efficiency and fairness.
>
> Thank you for the suggestions you provided. We plan to make the following changes in our revised manuscript:
>
> - Regarding the potential limitations, We completely agree with you that our proposed design is primarily tailored to optimize the experimenter's objective instead of the people who are enrolled in the experiment. To address this limitation, we will add the following discussion in our revised manuscript:
>    - "We acknowledge that our experimental design carries practical limitations. Despite its intention to safeguard enrolled subjects from unfair treatment allocation, the primary aim of this design remains more aligned with the experimenter's interests than those of the enrolled participants. This indicates a potential area for improvement in aligning the experiment objectives more with the participants' interests. "
> - We will incorporate the figures in the attached pdf file into our revised Supplementary Materials.
> -  We will add more discussions on the trade-off between the fairness and welfare constraints in our revised manuscript.
>
> Thank you again for reading our manuscript and for providing valuable suggestions.

---

### Official Review · Reviewer_ZNjb · 2023-07-11

**Soundness:** 4 excellent
**Presentation:** 4 excellent
**Contribution:** 3 good
**Rating:** 6
**Confidence:** 3

**Summary:**

This paper considers randomized experiments that aim to assess the effectiveness of some treatment or policy. The data and sample efficiency of such randomized experiments has been shown to improve by making them adaptive — i.e. by updating the treatment assignment probabilities on the fly, based on the data collected in the study. This paper argues that this can lead to fairness concerns, however. The goal of this paper is to strike a balance between maximizing information gain through randomized experiments and respecting fairness considerations. The fairness aspects considered in this paper include maximizing overall welfare of the participants (through treatments) and ensuring all participant groups in the experiment receive a “fair” exposure to the treatment.
Towards achieving this, the paper proposes a “fair adaptive experimental design strategy” that integrates fairness and welfare considerations along with optimizing information gain. The approach is to build a non-parametric algorithm that calculates the mean and variance of potential outcomes at the group level, each time step, to determine future allocations. The paper presents theoretical results that show that this approach yields strongly consistent estimates. Numerical results show that this approach improves upon the fairness considerations.


**Strengths:**

– I think the key contribution/strength of the paper lies in recognizing and attempting to fill the fairness gap that can arise in adaptive experiments.

– The problem formulation and contribution is cleanly spelled out and well-presented. Everything is easy to understand and follow.

– The paper presents nice theoretical results backing the soundness of the proposed approach.


**Weaknesses:**

– The experimental section appears a little weak. The DGPs used are completely synthetic and seem to be hand-crafted to demonstrate the utility of this approach. It may be more convincing to have more real/realistic datasets or even synthetic data generated from some real DGP (somewhat similar to the case study, but having one more domain might make it stronger).

– The abstract is either too dense or introduces too many technical terms that are discussed only later in the paper, making it very hard to understand. Perhaps cramming less info in the abstract might help.

– THe paper doesn’t discuss/ quantify the trade-off involved in achieving fairness: how much information gain is given up as compared to other SOTA adaptive experimentation approaches. Having some insights/ theoretical guarantees here may be very useful.


**Questions:**

– In section 2.1, is $n_t$ the number of *new* participants added to the experiment at time $t$ or simply the total number of participants at time $t$? If it is the former, then why is $D_{it}$ defined only for $i = 1, …, n_t$? If it is the latter, then N= \sum n_t doesn’t seem to make sense (because some participants may get over counted)? Could you please clarify this?

– Why is Assumption 1 mild / why is it reasonable or practicable to assume?

– In the statement of theorem 1, what is $n_1$ and do you need N tending to infinity? I thought that is already captured by the math? Or am I missing something?

– Could you elaborate on the tradeoff involved (data efficiency vs fairness) as mentioned under the last bullet under wekanesses?


**Limitations:**

No negative societal impacts discussed (but I don’t see any concerns).

Perhaps the technical limitation is that the approach may give up some amount of data efficiency. This could be discussed more in the paper.

---

> ### Author Rebuttal · Authors · 2023-08-06
>
> Thank you for your valuable feedback and your careful reading of our manuscript!
>
> - Regarding the definition of $n_t$, $n_t$ is indeed the number of new participants added to the experiment at time $t$. $D_{it}$ refers to subject $i$ enrolled at time $t$, therefore, $D_{it}$ ranges from $i=1,\ldots, n_t$, for $t=1,\ldots, T$. The total sample size can be expressed as $N=\sum_{t=1}^T \sum_{i=1}^{n_t} D_{it} = \sum_{t=1}^T n_t$.
> - As for Assumption 1, it assumes that the potential outcomes have finite $2+\delta$ momenets, which we employ in our theoretical analysis to establish, for example, the asymptotic normality of the treatment effect estimators. This assumption can be violated, for example, if one deals with data exhibiting heavy tails (such as in certain financial applications). In those cases, different mathematical tools are needed (such as stable convergence). In this manuscript, we follow the existing literature and make this assumption. Assuming potential outcomes with finite moments will be plausible if, for example, the outcome variables are naturally bounded. In our case study, the outcome variable is loan renewal, which is bounded. We also plan to add another case study in which the outcome variable is the length of stay in the hospital measured in hours, which is also a bounded outcome.
> - Regarding the statement of Theorem 1:
>    - $n_1$ represents the number of subjects enrolled in Stage 1. We require the total sample size $N$ tending to infinity.
>     -  In our previous manuscript, we require both $n_1$ and $N$ to go to infinity. However, upon further revision, we have found that the restriction on the size of $n_1$ can be dropped. Only having $N\rightarrow\infty$ would be sufficient for Theorem $1$ to hold.
> - There is indeed a trade-off between data efficiency vs. fairness. As our design objective is to minimize the variance of the estimated ATE under fairness constraints, we indeed sacrifice some data use efficiency while maintaining fair allocation among different subgroups. Such trade-offs can also be seen from Figure 2 and Figure 3 in our main manuscript. More concretely, we shall discuss the trade-off in different scenarios.
>    -  In our design,  $c_1$ in the envy-free constraint is a positive constant between $0$ and $1$. When an experimenter reduces $c_1$, a more stringent fair allocation rule is imposed and thus data use is less efficient, meaning that more subjects need to be enrolled in order to minimize the variance of the estimated ATE.
>    -  If $c_1$ equals $0$, our design resembles the completely randomized experiment, wherein treatment allocations are the same across different subgroups. However, this setup sacrifices data use efficiency, as it does not take into account possible variations in different subgroups.
>    - If $c_1$ equals $1$, our experimental design resembles the Neyman allocation, which prioritizes maximizing data use efficiency. However, under this setup, fair allocation across different subgroups is not guaranteed.
>
> - Regarding the DGP in our manuscript, we apologize if the description in our simulation study caused confusion. The DGP in our case study is indeed based on a real dataset (Banerjee et al., 2014). We first estimate useful parameters (such as group-level treatment effects and moments of the potential outcomes) from the real data, and then employ the estimated parameters to generate synthetic data for our case study.
>
> Per your suggestions, we plan to make the following updates to our manuscript.
>
> - We completely agree with you that adding an additional case study with applications in another domain would strengthen the manuscript. We have found a clinical dataset investigating the treatment effect of genetically-guided thearpy on treating major depressive disorder patients in Ruano et al. (2021). We hope to evaluate our design by generating synthetic data from this dataset in our revised manuscript.
>
>   - It is a suitable dataset because genetically-guided therapy is potentially beneficial for patients, therefore we hope to have fair treatment allocations in different subgroups. In addition, as patient subgroups may  respond differently to the therapy, it is natural to incorporate the welfare constraint. With this dataset, we hope to design adaptive experiments to efficiently estimate the treatment effect of genetically-guided therapy on MDD patients under fairness constraints. More details of the dataset are described as follows.
>
>
>    - The original trial is conducted at the Institute of Living at Hartford Hospital, consisting of 1459 patients (Tortora et al., 2020). There are two considered therapies: (1) the standard therapy ($D=0$), which does not rely on CYP2D6 functional status; (2) the genetically-guided therapy ($D=1$). The outcome is the length of stay in the hospital, measured by hours. The shorter the length of stay, the more beneficial the therapy is. The patient subgroups are defined by age: (1) 18-20; (2) 21-30; (3) 31-40; (4) 41-50; (5) 51-60; (6) $>60$.
>
> -  We will shorten the abstract in our revised manuscript.
>
> - We will add explanations on the trade-off between data use efficiency and fairness in our revised manuscript.
>
> We sincerely appreciate the questions you raised and the valuable suggestions you provided.
>
> Reference:
> 1. Banerjee, A., Duflo, E., and Hornbeck, R. (2014). Bundling health insurance and microfinance in
> india: There cannot be adverse selection if there is no demand. American Economic Review:
> Papers & Proceedings, 104(5):291–297
> 2. Ruano, G., Tortora, J., Robinson, S., Baker, S., Holford, T., Winokur, A., and Goethe, J. W.
> (2021). Subanalysis of the cyp-guides trial: Cyp2d6 functional stratification and operational
> timeline selection. Psychiatry Research, 297:113571.
> 3. Tortora, J., Robinson, S., Baker, S., and Ruano, G. (2020). Clinical database of the cyp-guides
> trial: An open data resource on psychiatric hospitalization for severe depression. Data in Brief,
> 30:105457.

---

### Author Rebuttal · Authors · 2023-08-10

We greatly appreciate all the valuable feedback provided by our reviewers.  In each rebuttal area, we have strived to make our best efforts to respond to the questions and comments raised by our reviewers.

To provide more evidence and comparison regarding the empirical performance of our design and also to answer some of the questions raised by our reviewers, we have attached a pdf file which includes the following three figures:

- Figure 1: We demonstrate the estimated optimal treatment assignment probability under our design compared to the oracle treatment assignment probability in finite samples.
- Figure 2: We compare our design with rerandomization regarding the estimation efficiency and the subgroup treatment allocation.
- Figure 3: We provide some insights on the estimation efficiency of our design with respect to various parameter values in the envy-freeness constraint to showcase the trade-off between fairness and efficiency.

Thank you again for your time in reviewing our manuscript!

---

### Decision · Program_Chairs · 2023-09-21

**Decision:**

Accept (poster)

**Comment:**

The authors introduce an interesting topic of study: adaptive experimental design with the goal of reducing estimation of group ATE variance, while maintaining allocation of treatment fairness constraints. Most reviewers are supportive of acceptance and I also investigated the paper and find the problem interesting.

I did find though lack of reference to fairness in contextual bandits papers and the authors are strongly encouraged to add such references if the paper gets accepted. See for instance:
Fairness in Learning: Classic and Contextual Bandits, https://arxiv.org/abs/1605.07139
and the many other papers that cite this work. It would be a great omission if this line of work is not contrasted. There are differences, which is why I still think it's ok to submit the current contribution: 1) most of that line of work focuses on regret as opposed to estimation of the ATE, 2) the fairness constraints are mathematically not the same.